# Eph Receptors in Cancer

**DOI:** 10.3390/biomedicines11020315

**Published:** 2023-01-23

**Authors:** Sakshi Arora, Andrew M. Scott, Peter W. Janes

**Affiliations:** Olivia Newton-John Cancer Research Institute, School of Cancer Medicine, La Trobe University, Heidelberg, VIC 3084, Australia

**Keywords:** Eph receptor, ephrin, receptor tyrosine kinase, cancer biology, targeted therapy

## Abstract

Eph receptor tyrosine kinases play critical functions during development, in the formation of tissue and organ borders, and the vascular and neural systems. Uniquely among tyrosine kinases, their activities are controlled by binding to membrane-bound ligands, called ephrins. Ephs and ephrins generally have a low expression in adults, functioning mainly in tissue homeostasis and plasticity, but are often overexpressed in cancers, where they are especially associated with undifferentiated or progenitor cells, and with tumour development, vasculature, and invasion. Mutations in Eph receptors also occur in various tumour types and are suspected to promote tumourigenesis. Ephs and ephrins have the capacity to operate as both tumour promoters and tumour suppressors, depending on the circumstances. They have been demonstrated to impact tumour cell proliferation, migration, and invasion in vitro, as well as tumour development, angiogenesis, and metastases in vivo, making them potential therapeutic targets. However, successful development of therapies will require detailed understanding of the opposing roles of Ephs in various cancers. In this review, we discuss the variations in Eph expression and functions in a variety of malignancies. We also describe the multiple strategies that are currently available to target them in tumours, including preclinical and clinical development.

## 1. Introduction of Eph Receptors

Erythropoietin-producing hepatoma (Eph) receptors represent the biggest family of receptor tyrosine kinases (RTKs), consisting of fourteen members that are split into two subcategories, nine A-type Ephs (EphA1–8, EphA10) and five B-type Ephs (EphB1–4, EphB6). Their distinction depends on their sequence similarity and ligand affinity, with EphAs preferentially binding five A-type ephrins (A1–A5) that are attached to the plasma membrane via a GPI-linkage, and EphBs binding three B-type ephrins (B1–B3) that are transmembrane proteins containing an intracellular domain [1,2]. This feature of binding to membrane-bound ligands, typically across cell–cell junctions, is unique among RTKs, and underlies their unique roles in controlling cellular interactions during normal and oncogenic development.

## 2. Structure and Signalling of Eph Receptors

The extracellular region (ECR) of Eph receptors consists of a ligand-binding domain (LBD), a cysteine-rich domain (CRD) including a sushi domain and an epidermal-growth-factor (EGF)-like domain, and lastly two fibronectin type III (FN3) domains, nFN3 and cFN3 (Figure 1). The LBD and CRD help the dimerisation and further clustering of receptors following initial ephrin binding [3]. The transmembrane domain creates a helix that links the extracellular region and intracellular region (ICR) by an axial insertion into the cell membrane. The ICR has a juxtamembrane (JM) domain, a tyrosine kinase domain, a sterile alpha motif (SAM), and a PDZ-binding domain. The JM domain is crucial for controlling kinase activity, interacting with its neighbouring kinase domain, and blocking substrate and nucleotide access to promote a dormant state, which is overcome by the phosphorylation of two conserved tyrosine residues in the JM area (JX1 and JX2). Mutation of these residues to phenylalanine eliminates the EphA4 kinase function, showing that the tyrosine phosphorylation of the JM region is essential to establish an active form [4]. Once exposed, phosphorylated tyrosine motifs in the JX and kinase domains also act as binding sites for proteins that have SH2 domains, linking to downstream signalling pathways [4,5].

A major impact of Eph receptor signalling is the modulation of the actin cytoskeleton via the Rho GTPase family, which includes RhoA, Rac1, and Cdc42, affecting cell morphology, adhesion, and motility [6]. Rac1 and Cdc42 stimulate the production of membrane protrusions such as lamellipodia and filopodia, while RhoA is primarily engaged in the development of stress fibres, focal adhesions, and the contraction of the actomyosin cytoskeleton [7]. GTPases bind downstream effectors in either their GDP-bound-dormant or GTP-bound-active form. Guanine nucleotide exchange factors (GEFs) and GTPase-activating proteins (GAPs) are two types of GTPase regulators that control switching between GDP and GTP bound states. [7]. The GEF ephexin is used by EphA receptors to activate GTPases and attaches to the kinase domain in cells of the nervous system [8]. The stimulation of EphA receptor proteins by ephrin-A results in ephexin-mediated RhoA activation, the inhibition of Cdc42 and Rac1, and modifications in the cell shape, which in turn triggers growth cone collapse [9]. The recruitment of Crk to ephrin-stimulated EphA3 in melanoma and 293T cells also caused a brief rise in activated Rho, which resulted in cell contraction and membrane blebbing [8,10].

The SAM domain has a role in mediating receptor dimerisation and downstream signal transmission. Phosphorylation of the conserved tyrosine residue Y928 in EphB1 and Y921 in EphA2 in the α2 helix of the SAM domain promotes the engagement of SH2-containing proteins such as Grb7 and Grb10 [11,12]. It has also been shown that the SAM domain of EphA2 can bind to the SAM domain of SH2-domain-containing inositol 5’-phosphatase 2 (SHIP2), inositol-polyphosphate-phosphatase-like protein 1 (INPPL1), and Odin (Anks1a) [13]. In addition, the SAM domain can control the function of the neighbouring kinase domain, since its removal from EphA2 increased tyrosine autophosphorylation in human prostate cancer cells and in a mouse skin carcinoma cell line, resulting in constitutive activity [9]. This may occur through increased clustering, as reported after the truncation of the SAM domain of EphA2 and EphB2, while, conversely, other research suggests that the inclusion of the SAM domain enhances EphA3 dimerisation in cells [12,14,15]. Lastly, Eph receptors possess a PDZ-binding motif at their C-terminus, recognised by proteins containing a PDZ domain, a 80–100 amino acid region named after the first three proteins found to include them: postsynaptic density protein of 95 kDa (PSD95), Drosophila disc large tumour suppressor (DlgA), and zonula occludens-1 protein (Zo-1) [16]. Ephs are reported to attach to PDZ-domain-containing proteins AF6 (a Ryk receptor tyrosine-kinase-interacting protein), Pick 1 (a protein-kinase-C-interacting protein), syntenin (a syndecan-interacting protein), and Grip1 and Grip2 (glutamate-receptor-interacting proteins), that are speculated to provide structural support for the construction of multiprotein, membrane-bound signalling complexes [17,18,19].

The canonical signalling mechanism by which Eph receptors carry out their functions includes ligand-induced clustering, tyrosine kinase activation, and adaptor protein binding. However, noncanonical signalling also occurs, encompassing low tyrosine kinase activity and serine phosphorylation of a linker region in both the KD and SAM domains. This phosphorylated linker then engages with adaptor proteins and downregulates the activity of protein kinase B, or Akt, and other Ser/Thr kinases that govern cell proliferation and viability via numerous downstream effectors, notably mTOR complex 1 [20,21]. Akt is normally activated by receptor tyrosine kinases (RTKs) via the lipid kinase PI3K (phosphatidyl inositol 3-kinase) by phosphorylating T308 and S473. Eph receptor forward signalling, on the other hand, can inhibit Akt activation [20]. Ephrin-dependent stimulation of EphA2 in several tumour cell lines causes rapid dephosphorylation of Akt T308 and S473, most probably via control of a phosphatase, which in turn inhibits mTORC1 and reduces cellular proliferation and motility [22,23,24]. Furthermore, the overexpressed inactive EphA2 is phosphorylated on Ser897 by Akt, which dramatically alters receptor activity, whereas Ser897 dephosphorylation is triggered by ephrin-A1 activation. Ligand-independent Ser897 phosphorylation of EphA2 stimulates migration/invasion and cancer progression, which is dependent on a decreased level of ligand-induced forward signalling [23]. 

Tyrosine kinase-dependent and -independent functions of EphBs have also been described. Blocking the kinase activity of three neuronally expressed EphBs in triple knock-in mice showed that EphB1, B2, and B3 kinase activity was not necessary for synapse formation, whereas it was required for ephrin-B-mediated growth cone collapse in vitro and the guidance of retinal and corpus callosal axons in vivo [25]. Interestingly, the levels of receptor/ligand expression, receptor clustering, and kinase activity are critical to determining physiological responses. Thus, high levels of receptor expression and ligand-stimulated phosphorylation can cause cytoskeletal collapse and cell or axon retraction, while lower-level expression/activation can cause the opposite response of adhesion, cell spreading, and axon extension [10,26].

## 3. Bidirectional Signalling

As mentioned above, the phenomenon of bidirectional signalling from Eph–ephrin engagements across cell–cell junctions is one of the characteristics that sets Ephs apart from other receptor tyrosine kinases [27]. While ephrin binding to Eph receptors on an adjacent cell causes receptor oligomerisation, transphosphorylation, and ‘forward signalling’, the simultaneous clustering of ephrins on the opposing cell membrane also sends a signal into ephrin-bearing cells. This is called reverse signalling [28]. This occurs either via signalling of the ephrin intracellular domain (only present in ephrin-Bs) or via protein interactions mediated by other membrane-anchored effectors, including lipid–protein interactions occurring in membrane microdomains in which ephrins are known to reside [10]. Both forward and reverse signalling are engaged in many essential physiological functions in humans, including the development of the nervous and vascular systems, tissue boundary formation, and tissue homeostasis [29,30]. While this review focuses on Eph receptors, the functions of their cognate ligands are integrally linked, and they are similarly implicated in the dysregulation of essential signalling pathways during tumour development, advancement, and metastasis [29].

## 4. Normal Function of Eph Receptors in Development and Adult Tissues

Eph receptors and ephrins are implicated in a broad variety of developmental activities, including cardiovascular and skeletal formation, axon guidance, and tissue patterning, and are found in almost all tissues of a growing embryo [31]. In gastrulation, somitogenesis, and in the establishment of tissue and organ boundaries, they direct cell migration and adhesion [32]. Throughout embryogenesis, Ephs and ephrins are expressed in complimentary regions, and their bidirectional signalling establishes borders across zebrafish rhombomeres, as well as in in vitro models of cell–cell segregation employing zebrafish blastomeres and mammalian cell cocultures of Eph- and ephrin-expressing cells [10]. The growing nervous system has the highest levels of expression for both Eph receptors and ephrins, and one of the primary developmental roles of these molecules is to control how and where new axons form [33]. Tiny subsets of neurons in the superior colliculus, hindbrain, and spinal cord are the only locations where EphA8 is expressed in the central nervous system. EphA8-null mice have a defect in which axons from a subset of neurons in the superior colliculus that typically travel to the contralateral inferior colliculus instead project to the ipsilateral cervical spinal cord. Mice lacking EphA4 have motor impairment, perhaps due to damage to the corticospinal tract. Furthermore, the front junction is absent in the vast number of these animals [34]. Evaluation of EphB receptor mutant mice demonstrated the need for EphB receptor signalling in spine formation in vivo. Spine density in EphB1/EphB2/EphB3 triple-null mice was much lower, with unusually tiny spines lacking heads. Knock-in mice producing a mutant form of EphB2 in which the kinase domain had been swapped by lacZ on an EphB1-deficient setting also exhibited impaired spine growth. These findings are supported by in vitro experiments showing that forward signalling from EphB receptors is necessary for appropriate dendritic spine development [33].

During angiogenesis, ephrins and Eph receptors are critical in defining the vascular-arterial boundaries. In the early phases of angiogenesis, arterial and vein endothelial cells may be differentiated from one another by their preferential expression of ephrin-B2 or EphB receptors. Ephrin-B2-deficient mice and EphB2/EphB3-double-deficient mice both have impaired angiogenesis and die in utero during gestation due to abnormalities in the remodelling of the embryonic vascular system [35,36]. Ephs are also known to have several functions in the formation, transportation, and stimulation of immune cells (both innate and adaptive). The attachment, secretion, and transportation of platelets, monocytes, macrophages, and dendritic cells (DCs) are all regulated by Ephs, as are the motility and activation of B and T lymphocytes [32]. Deployment and maturation of hematopoietic stem cells (HSCs) are controlled by engagement between HSCs and bone marrow stromal cells, which are mediated by EphB2/4 and ephrin-B2 [32].

Physiological roles of Ephs and ephrins in adult tissues are still being defined. Although generally their expression is downregulated in adult tissues compared to during embryogenesis, they still function in adults, including roles in stem cell homoeostasis, and in preserving the plasticity and regeneration potential of adult tissues and organs. Emerging research has linked them to cognitive processes including learning and memory, as well as to bone maintenance and insulin production [31]. Eph–ephrin interactions are also critical for the regulation of vascular and bone restructuring, the regulation of stem cell placement and growth, and synaptic remodelling in reaction to brain and central nervous system damage [37]. In the adult brain, Eph receptor proteins are expressed in areas where rewiring of nerves is still taking place, such as the cerebral cortex, the hippocampus, the cerebellum, and the olfactory bulbs. They are also highly enriched and critical in synapses [33]. Similarly, Ephs are involved in angiogenesis in adult tissues. EphA3 is upregulated by hypoxic signalling in mesenchymal cells contributing to the neovascularisation of the regenerating human endometrium during the menstrual cycle, but not in other highly vascularised human organs [38]. Lastly, functions of Ephs also re-emerge in diseases, including atherosclerosis and fibrosis, and most notably cancer, as described below [32].

## 5. Ephs in Cancer

Ephs have been the subject of considerable research regarding their functions in tumour development. Due to their generally low expression in mature tissues and elevated expression in numerous cancers, Ephs have garnered the greatest interest as tumour neoantigens in tumour immunology. EphA3 was the first Eph receptor recognized as a tumour-associated antigen (TAA) [27]. It was originally identified using an antibody raised against a lymphoblastic leukemia cell line [39]. Independently, a CD4+ T-cell clone derived from a melanoma patient with disease regression was found to recognise an EphA3 epitope and to induce a preferential immune response against melanoma cells [40]. Subsequently, a range of Eph receptors have been identified as preferentially expressed on tumours, of which EphA2, EphA3, and EphB4 have been a particular focus for therapeutic targeting [41]. They have been implicated in the formation of a variety of malignant neoplasms, including lung, prostate, colon, pancreatic, ovarian, thyroid, tongue, and hepatocellular carcinomas, as well as gliomas and melanomas [1,2,30,42]. However, their functions vary, and, as described below, they can have both tumour-suppressing and tumour-promoting functions in different contexts (Figure 2, Table 1).

### 5.1. Tumour-Promoting Function

EphA3 was shown to have an oncogenic role in GBM tumours. It is very weakly expressed in the healthy brain but is abundantly expressed on glioblastoma stem cells (GSCs), where it plays an important role in regeneration and the long-term survival. Tumour cell differentiation and apoptosis were seen following EphA3 knockdown in vivo [43]. Levels of EphA2, EphB2, and EphB4 are reported to be higher in human breast tissue than in normal mammary epithelial cells. While the EphB2 protein was detected in all normal tissue samples, it was shown to be overexpressed in 51% of breast tumours, and patients with higher EphB2 expression had a worse prognosis. EphB4 protein expression was also associated with increasing stage and histological grade, and cell proliferation and DNA aneuploidy were both linked to a rise in EphB4 membrane staining [44]. Small-interfering-RNA (siRNA)-mediated knockdown of EphB4 expression resulted in a dose-dependent decrease in cell survival, increased apoptosis, and a heightened sensitivity to the tumour-necrosis-factor-related apoptosis-inducing ligand (TRAIL) in breast cancer [45]. EphA2 is also overexpressed and associated with poor prognosis in breast cancer, where it amplifies oncogenic signalling of the RTK erbB2 (HER2), and the loss of EphA2 in the mammary epithelium of mice slowed down tumour development and metastasis [46]. In a mouse model of pancreatic adenocarcinoma, decreased EphA2 expression using siRNA suppressed tumour development and progression, including invasion and metastasis [47]. An increased expression of EphA4, EphA7 and EphA10 receptors was also found in breast cancer, which is associated with poor prognosis, and EphA4 is reported to promote breast cancer cell proliferation, migration, and invasion via the transforming growth factor-beta (TGFβ) signalling pathway. These findings, in addition to examples for specific tumour types discussed later, suggest that Eph receptors can play a crucial role in tumour promotion.

**Figure 2 biomedicines-11-00315-f002:**
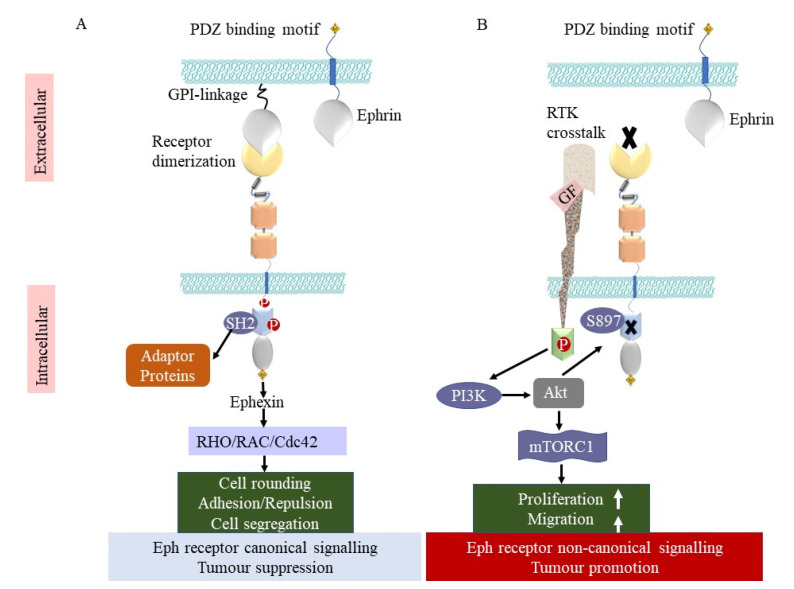
Tumour-suppressing and tumour-promoting functions of Eph receptors. (**A**) Interaction of Eph receptor and ephrin on neighbouring cells causes clustering of Eph–ephrin complexes and the beginning of canonical signalling. The activation of Eph kinase function involves tyrosine phosphorylation and recruitment of adaptor proteins, including SH2-domain-containing signalling proteins. In reaction to ephrin-A, ephexin stimulates RhoA, which induces cytoskeletal remodelling, cell retraction and segregation, and membrane blebbing. (**B**) In the absence of ephrin, Eph receptor expression can lead to noncanonical signalling, whereby crosstalk between Eph receptors and other RTK activity induces phosphorylation of Akt by the lipid kinase PI3K. Akt then phosphorylates EphA2 at S897 to enhance cellular proliferation, cell migration, and reduce apoptosis through several downstream effectors, including mTOR complex 1.

### 5.2. Tumour-Suppressing Function

Some tumour tissues have also been shown to have lower Eph or ephrin levels compared to normal tissues. For instance, EphA1 is underexpressed in aggressive colon cancer which is associated with a poor prognosis in patients [48]. EphB4 and ephrin-A5 are significantly suppressed in colorectal and glioblastomas, respectively. Advanced lung tumours had lower levels of EphB6 expression [49]. After being activated by its ligand ephrin-B2, the EphB4 receptor was shown to have tumour-suppressing effects in mice xenografts of breast cancer. EphB4 induces antioncogenic signalling in breast cancer cells that involves Abl tyrosine kinase and the Crk adaptor protein. These findings imply that EphB4 acts as a tumour suppressor when triggered by its ligand, and that the tumour-promoting actions of Eph receptors may be ligand-independent in this setting, where the Abl–Crk pathway suppresses the growth, migration, and infiltration of breast cancer cells [6]. Consistent with this, high ephrin-B2 expression in breast tumours was associated with lower grade tumours and better patient prognosis, and its expression in vitro inhibited proliferation and migration [50].

EphBs are also associated with gastrointestinal tumour suppression. EphB4 is a putative tumour suppressor gene in colorectal cancer, since its expression is commonly decreased or deleted [51]. In a mouse xenograft model, decreasing EphB4 expression in colon cancer cells led to increased tumour growth [52]. Further, the loss of a singular allele of EphB4 in a genetic model of intestinal cancer with adenomatous polyposis coli (Apc) mutation resulted in greater epithelial proliferation and bigger tumours in the small intestine [52]. Similarly, EphB2 and EphB3 deficiency in Apc^Min/+^ mice increased the frequency, size, and aggressiveness of colon tumours [53]. Subsequent studies showed that that ephrin-B1 controls the distribution and spread of EphB2- and EphB3-expressing tumour cells in the colon, which is overcome during tumour progression by the loss of EphB expression [54].

**Table 1 biomedicines-11-00315-t001:** Expression of Eph receptors in cancers and association with tumour progression and patient prognosis.

Receptor	Cancer Type	Upregulated/ Downregulated	Tumour Promoting/Tumour Suppressing	mRNA/ Protein	Prognosis	Reference
EphA1	Colorectal	Downregulated	Tumour suppressing	Both	Poor survival	[48]
EphA2	Breast Pancreatic	Upregulated Upregulated	Tumour promoting Tumour promoting	Both mRNA	Poor survival Poor survival	[55,56,57]
EphA3	Brain Colorectal Prostate	Upregulated Upregulated Upregulated	Tumour promoting Tumour promoting Tumour promoting	Both Protein Protein	Poor survival Poor survival Poor survival	[42,58]
EphA4	Lung Breast	Upregulated Upregulated	Tumour suppressing Tumour promoting	Both mRNA	Increased survival Poor survival	[56,59]
EphA7	Breast	Upregulated	Tumour promoting	mRNA	Poor survival	[56]
EphA10	Breast	Upregulated	Tumour promoting	Protein	Poor survival	[60]
EphB2	Breast Colorectal	Upregulated Downregulated	Tumour promoting Tumour suppressing	Protein mRNA	Poor survival Poor survival	[44,53]
EphB3	Colorectal	Downregulated	Tumour suppressing	Both	Poor survival	[53,61]
EphB4	Colorectal Breast	Downregulated Upregulated	Tumour suppressing Tumour promoting	Protein Both	Poor survival Poor survival	[44,51,56]
EphB6	Breast	Upregulated	Tumour promoting	mRNA	Poor survival	[56]
Ephrin-B2	Breast	Upregulated	Tumour suppressing	Protein	Increased survival	[50]

Although EphA2 has been implicated in cancer, there are also studies suggesting it may have antitumorigenic roles. Mice lacking EphA2 were shown to be more susceptible to chemically induced skin carcinogenesis, leading to increased tumour growth and invasion [62]. EphA4 expression was found to be increased in breast and lung cancer compared to normal tissues [56]. Expression in lung cancer was linked to better prognosis in patients following tumour resection. In vitro upregulation of EphA4 decreased cell penetration and motility, while having no effect on drug susceptibility, cell cycle, and apoptosis, suggesting EphA4 may influence tumour cell migration and invasion [59]. Together, studies show the complex roles of Ephs in cancer, where they can be associated with either tumour-promoting or supressing roles, dependent on receptor/ligand expression, kinase activity, and signalling crosstalk.

### 5.3. Mutations

Mutations in the genes encoding Eph receptors have been found in the screening of tumour samples and cell lines and some are thought to be involved in cancer development. Tumours of the human prostate, stomach, colon, and melanoma have all been shown to include alterations in EphB2 [7]. It is possible that these mutations impede kinase activity and that some of them occur in tandem with losing heterozygosity. EphB2 is situated on the region of chromosome 1, called p35–36, and has been suggested as a potential tumour suppressor gene because of its association with recurrent allelic inactivation in colorectal cancers [44]. Several Eph receptors, including EphA5 and most commonly EphA3, are mutated in lung cancer [7], and EphA3 mutations have also been reported in melanoma, glioblastoma, colon, liver, pancreatic, and ovarian cancers [63]. Inhibition of tumour growth in vivo by two NSCLC cell lines expressing wild-type EphA3, but not EphA3 mutants, shows that the EphA3 can act as a tumour suppressor in lung cancer [64]. Accordingly, reduced kinase activity or tyrosine phosphorylation was seen for EphA3 alterations in NSCLC, and no alterations resulted in elevated action [64]. Together, these studies suggest that, at least in some situations, elevated levels of mutated Eph receptors with reduced activity can promote tumour development.

### 5.4. Roles in the Tumour Microenvironment

The overexpression of Ephs and ephrins is not only seen in cancer cells but also in the tumour microenvironment (TME) (Figure 3). Since EphA2 is involved in vascular remodelling, and highly expressed in vascularised tumours, Chen et al. investigated its role in 4T1 mouse mammary tumours using EphA2 null mice. They showed that a lack of EphA2 in the tumour microenvironment, notably in the blood vessel endothelium, hinders tumour angiogenesis and metastases. Tumours derived from EphA2 null mice showed a substantial reduction in tumour volume compared to wild-type mice and an increase in tumour cell death [55]. Ephs have also been identified on other cell types that can be recruited to tumours and promote tumour survival, such as tumour-associated macrophages (TAMs), myeloid-derived suppressor cells (MDSCs), and mesenchymal stromal cells (MSCs). These cell types can also promote angiogenesis, as well as inhibit the function of T lymphocytes, thereby limiting their ability to kill cancer cells [32]. Vail et al. discovered EphA3 upregulation in the microenvironment of a range of solid tumour types, as well as in prostate and colon xenografts, where it was expressed on MSCs and myeloid cells recruited from the bone marrow. Treatment with an agonistic antibody caused the retraction of EphA3^+^ stromal cells in vitro and the disruption of the tumour stroma and vasculature in vivo, with a corresponding decrease in tumour growth. [42].

Eph-mediated interaction between cancer stem cells (CSCs) and TAMs has also been described in breast cancer, mediated by EphA4 [65]. EphA4 was elevated following epithelial to mesenchymal transformation of mammary epithelial cells, which gives tumour cells stem-cell-like traits that are linked to aggressiveness, invasion, and resistance to treatment. Breast CSCs expressing EphA4 and the MSC/CSC marker CD90, which binds integrins on surrounding cells, were evident at tumour margins in mice, interacting with invading TAMs [65]. The TAMs caused EphA4 activation and stimulated cytokine secretion in the CSC population, resulting in enhanced tumour cell proliferation via Src kinase, phospholipase Cγ1, protein kinase C, nuclear factor kappa B, IL-6, and IL-8 activation [65]. In pancreatic cancer, EphA2 activity on tumour cells regulated immunological suppression by excluding T-cells. EphA2 was shown to have the highest expression level among Ephs in pancreatic tumours, and its expression correlated with a loss of T-cell infiltration. In a K-Ras mutant mouse model, knockout (KO) of EphA2 increased the number of CD8+ and CD4+ T-cells in tumours, while decreasing the number of immunosuppressive MDSCs and TAMs [57]. EphA2KO cancer cells in tumours were more responsive to treatment with both chemotherapy and immunotherapy. Elevated IFN sensitivity and inflammatory pathways were identified in EphA2KO tumours as the basis for the enhanced immune reaction, while they had decreased activity of TGF/SMAD signalling and the downstream effector Ptgs2/cyclooxygenase-2 (COX-2) [57]. EphA10 has also been shown to suppress T-cell-mediated death in syngeneic mammary tumours via increased PD-L1 expression [66]. An RTK array of EphA2, EphA4, and EphA10 revealed that all three Ephs were tyrosine phosphorylated, including the kinase-dead EphA10, which may have been crossactivated by another Eph [66]. Only EphA10 was necessary for PD-L1 overexpression and the removal of EphA10 from mouse 4T1 cancer cells greatly enhanced CD8+ T-cells, T-cell activity, and tumour cell death, and decreased tumour growth [66].

### 5.5. Mechanisms of Drug Resistance Mediated by Eph Receptors

Preclinical and clinical studies show that resistance to cancer therapies often occurs following an initial period of response. Eph receptors are implicated in enhancing drug resistance via modulating other cancer pathways. EphA2 overexpression in breast cancer is associated with poor prognosis and has been implicated in mechanisms of resistance to EGFR family inhibitors, as recently reviewed [67]. Upregulation of EphA2 reduced dependence on the oestrogen receptor function, hence diminishing tamoxifen’s capacity to suppress breast cancer cell proliferation and carcinogenesis [68]. Furthermore, the anti-erbB2 antibody trastuzumab increased the phosphorylation of EphA2 by stimulating Src kinase, which in turn promotes signalling via the PI3K/Akt and MAPK pathways, resulting in resistance to trastuzumab [69]. EphA2 is also a driver of resistance to the BRAF inhibitor vemurafenib, where resistant cells displayed a more motile and invasive phenotype dependent on EphA2. Direct inhibition of EphA2 was shown to reduce Akt and erk (MAP kinase) phosphorylation, induce apoptosis, and efficiently reduce melanoma development in vivo [70]. EphA4 engages with cyclin-dependent kinase 5 (CDK5) in multiple myeloma (MM) and increases its expression, and facilitates bortezomib resistance by increasing Akt phosphorylation [71]. EphB4–ephrin-B2 interaction had also been involved in resistance of chronic myeloid leukaemia to the Abl kinase inhibitor imatinib, which was mitigated by inhibiting EphB4 receptor expression. EphB4 knockdown prevented cell motility and recovered imatinib susceptibility in vivo and in vitro, accompanied by elevated levels of phospho-EphB4 and lower levels of RhoA, Rac1, and Cdc42 phosphorylation [72]. Melanomas with EphB4 upregulation were also more resistant to cisplatin chemotherapy and showed elevated levels of phospho-Akt and phospho-Erk, which was related to resistance. Accordingly, EphB4 inhibition restored Cisplatin sensitivity [73]. EphA3 reduced expression has been linked to the PI3K/BMX/STAT3 pathway, which has been found to cause multidrug resistance. EphA3 upregulation in small-cell lung cancer (SCLC) lowered therapeutic resistance by promoting apoptosis and triggering G0/G1 arrest, which was associated with decreased phosphorylation of the PI3K/BMX/STAT3 signalling [74]. These findings show that the Eph receptors have a role in multiple mechanisms of the drug resistance of tumours.

### 5.6. Eph Receptors as Therapeutic Targets in Specific Cancer Types

As stated above, the roles of Eph receptors and their ephrin ligands have been identified in a range of different tumours. Examples of some of the major tumour types are summarised below.

#### 5.6.1. Lung Cancer

Lung cancer kills more people than colorectal, breast, and prostate cancers put together, making it the top cause of cancer fatalities around the world. Nonsmall-cell lung cancer (NSCLC) makes up about 80% of all lung cancers. Brannan et al. showed that the upregulation of EphA2 in NSCLC is linked to a poor prognosis and the emergence of K-Ras mutations. Knocking down EphA2 levels slowed the proliferation and motility of NSCLC cells in culture and higher EphA2 expression was linked to metastases in NSCLC [75]. Amato et al. showed that the disruption of EphA2 in a mouse model of invasive K-Ras mutant NSCLC suppressed tumour development. In human NSCLC cell lines, EphA2 knockdown decreased cell viability and proliferation. EphA2 inhibition reduced phosphorylation of apoptotic agonist BAD and caused apoptosis in NSCLC tumours in mice, blocking tumour growth [76]. These studies showed that EphA2 increases NSCLC development and can be a therapeutic target. EphB3 was also shown to be more abundant in NSCLC samples than in normal tissues, and its expression was associated with tumour growth and metastasis. Overexpression of EphB3 in NSCLC cell lines accelerated cell growth and migration and promoted tumorigenicity in xenografts in a kinase-independent manner, supporting noncanonical Eph function in tumorigenesis. In contrast, the downregulation of EphB3 inhibited cell proliferation and migration and suppressed in vivo tumour growth and metastasis. [77]. This suggests that EphB3 could also function in the development of NSCLC.

#### 5.6.2. Brain Cancer

Glioblastoma (GBM) is by far the most common form of brain cancer. GBM cells with stem-cell-like properties are very resilient to chemotherapeutics and radiotherapy and may regenerate tumours following treatment, contributing to the disease’s extremely aggressive character [78]. Binda et al. discovered that human GBMs (hGBMs) had 100-fold greater EphA2 mRNA expression compared to regular brain tissue. EphA2-high hGBMs were more tumourigenic, since mice injected with these cells died sooner than those injected with EphA2-low cells. Interestingly, treatment in vitro with ligand ephrin-A1-Fc diminished their ability to proliferate and form steady TPC lines [79]. A separate study showed EphA2 is present in glioma stem cells, wherein it enables ligand-dependent signalling facilitated by ephrin-A1 via Akt and ERK suppression, and ligand-independent Akt signalling via phospho-S897, which was inhibited by the ligand. EphA2 upregulation promoted intracranial penetration, while ephrin-A1/A3/A4 triple-knockout (TKO) mice exhibited enhanced GSC invasion compared to the wild-type control [80]. These studies demonstrated that ligand-independent EphA2 signalling is crucial for the pathogenesis of hGBMs, which can be counteracted by ligand expression. Similarly, Day et al. found elevated levels of EphA3 expression in GBM, which was more prominent on less differentiated tumour cells that coexpressed integrin α6, a marker of stem-like cells. In orthotopic mouse GBM xenografts, treatment with radiolabelled EphA3 monoclonal antibody (mAb) IIIA4 blocked tumour growth, supporting the utility of EphA3 as a therapeutic target in GBM [58]. Furthermore, Qazi et al. reported that recurrent GBM (rGBM) have increased expression of both EphA2 and EphA3 and demonstrated that their coexpression is linked to strong tumorigenicity in vitro and in vivo. Combined EphA2 and EphA3 knockdown inhibited the clonogenic ability of rGBMs, increased apoptosis, and reduced GSC markers, indicating the elimination of the undifferentiated stem-like cells. A bispecific antibody (BsAb) targeting of EphA2 and EphA3 inhibited rGBM xenografts in mice and reduced in EphA2 and EphA3 expression [81]. EphB2 is also expressed in GBM and was reported to promote GBM neurosphere invasion and migration via focal adhesion kinase (FAK) signalling while inhibiting neurosphere cell proliferation. Suppressing EphB2 also increased the invasiveness of EphB2-overexpressing GBM neurosphere xenografts in mice, suggesting that GBM invasion may be targeted by blocking EphB2 signalling [82]. 

#### 5.6.3. Gastrointestinal Cancers

Ephs and ephrins have been shown to have a crucial role in gastrointestinal cancers. While, as mentioned previously, the loss of EphB receptor expression was connected with the invasion of colorectal cancer [53], they have also been shown to promote colon stem cell proliferation and adenoma formation [83,84]. Other studies support the protumour roles of Ephs in the colon. Lv and colleagues found that CRC cell lines manipulated to overexpress EphB4 grew faster as xenografts and had a more vascular and invasive morphology [85]. High EphA1 expression has also been observed in the early stages of CRC, while low levels were more common in later stages and predicted a worse outcome [48]. EphA2 and ephrin-A1 increase was also more prevalent in the initial phase of cancer development as compared to the later phase, and ephrin-A1 overexpression promoted the proliferation of HT29 colorectal cancer cells [86]. High EphA3 expression was also linked to increased tumour volume, grade, and metastases and a much worse prognosis in hepatocellular carcinoma (HCC). HCC cell invasiveness was inhibited in vitro by VEGF modulation, which was mediated by the suppression of EphA3, suggesting potential as a prognostic marker and target for HCC treatment [87].

#### 5.6.4. Breast Cancer

Breast cancer (BC) is the most common aggressive tumour among females. Several studies have examined the involvement of the Eph receptors in BC, as recently reviewed by Psilopatis and colleagues [88]. EphA2 and EphB4 have been the most extensively studied in relation to breast cancer, although other Eph receptors have also been identified. EphA2 is highly expressed in 40 percent of breast malignancies and is often connected with a worse prognosis. This upregulation has been associated with mammary epithelial cell transition, driving cancer cell movement in vitro and triggering tumour growth when injected into nude mice [89]. As discussed earlier, EphA2 also amplifies the oncogenic signalling of the RTK erbB2 (HER2), and the loss of EphA2 in the mammary epithelium of mice slowed down tumour development and metastasis [46]. The upregulation of EphA4 and EphA7 receptors in breast cancer was also associated with poor survival. Transforming growth factor-beta (TGFβ) signalling from EphA4 has been linked to breast cancer cell growth, motility, and penetration. Lymph node metastasis in breast cancer was also linked to EphA10 expression. Since EphA10 is a kinase-deficient receptor, it has been postulated that it executes its functions via an association with EphA7 [90].

#### 5.6.5. Prostate Cancer

Therapeutic approaches that promote ligand-like signalling may be especially useful in the treatment of prostate malignancies, where the upregulation of Eph receptors lead to the initiation of ligand-independent signalling. In vitro studies using several human PCa cell lines such as PC3, LNCaP, DU145, and 22Rv1 have revealed that ligand-independent pathways, which encourage carcinogenic and aggressive behaviours, are activated with a high expression of receptors such as EphA2 and EphB4. This is reversed by stimulating overexpressed EphB4 with the soluble ephrin-B2 ligand, demonstrating the kinase-dependent tumour suppressor properties [91]. EphA2 expression was shown to be elevated in prostatic intraepithelial neoplasia, the predecessor to prostatic adenocarcinoma, suggesting a potential function for EphA2 in the early phases of prostatic tumorigenesis [89]. EphA3 was also overexpressed in aggressive cell lines, suggesting a role in malignancy [92]. EphA3 was expressed in the stromal and vascular tissues of human tumours and prostate mouse xenografts, where its treatment with anti-EphA3 mAb IIIA4 inhibited tumour growth, suggesting EphA3 as a novel target for the selective targeting of the tumour microenvironment [42]. EphA4 expression is also related to a more aggressive phenotype, and its significance has been emphasised by siRNA silencing, which led to a decrease in the viability of prostate cancer cells [93]. Interestingly, the knockdown of ERBB3/HER3, a receptor linked with prostate cancer, in DU-145 cells culminated in EphA4 downregulation, suggesting that ERBB3/HER3 regulates EphA4 levels, as shown by Soler et al. [94].

#### 5.6.6. Melanoma

Research has shown roles for Ephs in melanoma. EphA2 expression is greater in metastatic cells than in initial melanoma cells, and Udayakumar et al. showed that eliminating EphA2 specifically from high-expressing melanoma cells caused substantial decreases in cell viability, colony formation, and migration in vitro and tumorigenicity in vivo, indicating EphA2 is a crucial survival factor in melanoma cells [95]. As described above, EphA2 is a mediator of resistance to vemurafenib and it has been shown that suppressing EphA2 reduces Akt/erk phosphorylation and inhibits melanoma progression in mice [70]. In other studies, the expression of the EphA2 ligand ephrin-A1 was also elevated in melanoma in 67% of metastatic melanomas and 43% of progressed primary melanomas, where the expression was enhanced by inflammatory cytokines TNF-α and IL-1β, and it was postulated to play a role in tumour angiogenesis via interaction with EphA2 on endothelial cells [8,96].

## 6. EphA10 and EphB6 (Pseudokinases)

EphB6 and EphA10 share the same general architecture as the rest of the Eph RTK family members but are catalytically dysfunctional because of alterations in key residues that are essential for their tyrosine kinase activities [97,98]. Although their precise roles in controlling Eph receptor signalling are unclear, it is likely that both EphA10 and EphB6 have noncatalytic regulatory roles. Surprisingly, the JM domain of EphA10 has the two conserved tyrosine residues (JX1 and JX2) replaced with phenylalanine and cysteine, while EphB6 retains these tyrosine residues [4]. Irregular expression of these proteins has been linked to tumourigenesis and indicates important functions in signal transduction. EphB6 expression was discovered to be lower in colorectal cancers in contrast to adenoma and healthy tissues, and its loss promotes tumour progression, since EphB6 knockdown elevated lung metastasis in mice, whereas reintroducing EphB6 into colon cancer cells substantially lowered metastases [99,100]. In contrast, EphB6 can promote cell growth in triple-negative breast cancer cell lines [101]. EphB6 has been found to engage with EphA2, EphB1, and EphB4, suggesting it may produce heterodimers and oligomers with these other Eph proteins in the plasma membrane [101]. EphA10 is upregulated in malignant cells, particularly those of the breast and lungs, and reducing its expression in the breast cancer cell line ZR-75-1 resulted in increased apoptosis [102]. Li et al. reported EphA10 upregulation and decreased expression of a soluble isoform was associated with increased breast cancer cell cancer invasion and spread via E-cadherin and β-catenin. Cellular infiltration and lymph node metastases were both suppressed when the normal isoform expression profile was restored [103]. These examples further emphasise the kinase-independent roles of Ephs in promoting cancer.

## 7. Therapeutic Strategies to Target Eph Receptors

Given their expression and functions in cancers, described above, the Eph/ephrin family has been the focus of various strategies for developing potential cancer therapies. These include small-molecule inhibitors, synthetic peptides that block Eph–ephrin binding, kinase inhibitors, and therapeutic mAbs [37] (Figure 4, Table 2).

### 7.1. Small-Molecule Inhibitors

A variety of small-molecule inhibitors preventing Eph–ephrin interactions have been identified that might serve as the foundation for new therapeutics. Derivatives of lithocholic acid, 2,5-dimethylpyrrolyl benzoate, and salicylate compete with ephrin-As for attachment to EphA receptors, preventing activity and cell rounding [37,104,105,106]. A Pseudomonas aeruginosa electron transfer protein called azurin inhibits ephrin interaction to EphB2 and interferes with upstream signalling, which slowed down the proliferation of prostate cancer cells [107]. Compound 76D10, a disalicylic acid–furanyl derivative, was discovered by Noberini et al. along with two isomers of the compound 2,5-dimethylpyrrolyl benzoic acid. All three of these compounds work to prevent ephrin-A5 from interacting with the EphA4 in HT22 neuronal cells, with micromolar IC50 values [106]. The effectiveness of compound **1** in vivo was subsequently confirmed in pancreatic xenograft mice models, in which it was shown to suppress the phosphorylation of EphA4 and Akt, resulting in apoptosis [108]. D5-cholenoyl-amino-acid derivatives are another kind of Eph inhibitor now under investigation. UniPR1331 (compound **10**) is a derivative that broadly inhibits ephrin ligand attachment to EphA and EphB receptors with IC50 values between 2.5 and 5.4 nM and inhibits kinase activation in vitro [109]. Sanguinarine, a natural benzophenanthridine alkaloid, has been demonstrated to downregulate hypoxia-induced pathways and inhibited tumour growth in BC xenografts [110].

### 7.2. Kinase Inhibitors

Eph receptor kinase inhibitors have been sought for in several different ways. By screening a combination of antagonists intended to link to the dormant kinase configuration (class 2 antagonists) in a cell-based assay assessing the phosphorylation of a chimeric EphB2, many drugs with strong affinity for the Eph family were found [111]. Inhibitors of EphB3 kinase activity were found in a collection of imidazo[1,2-a] pyrimidines and pyrazolo[1,5-a] pyridines using a high-throughput screen. These compounds attack tyrosine kinases rather than serine/threonine kinases [112]. However, the majority of attention has been paid to the EphB4 receptor because of its role in angiogenesis. Crystal-structure-guided refinement helped narrow down the pool of potential kinase antagonists to a few distinct families, including 2,4-bis-anilinopyrimidine compounds [113]. NVP-BHG712 is an antagonist that was found by the computational modelling of the EphB4 kinase domain, then optimised for blocking of EphB4 phosphorylation in cells. It has a long half-life and displays a high affinity for EphB4. After oral treatment, it blocks VEGF-driven revascularisation and suppresses phosphorylation of EphB4 in tissues [114].

Dasatinib is a kinase antagonist that blocks the activity of c-KIT, PDGFR, and SFKs, used for treatment of leukaemias (CML, ALL). Multiple investigations have shown that dasatinib also directly inhibits EphA2 kinase activity and phosphorylation [115]. Recently, dasatinib has also served as a basis for improving EphA2 inhibitors. Along with eicosapentaenoic acid, it is responsible for building up of ATP-binding-cassette-subfamily-A-member-1 (ABCA1)-dependent cholesterol, which increases the polarity of the plasma membrane and promotes apoptosis in triple-negative breast cancer (TNBC) cells [116]. The new dasatinib-derived EphA2 inhibitor, compound 4a, was demonstrated to have improved specificity while retaining significant inhibitory actions toward EphA2 and proliferation in glioblastoma cells [117]. Kinase inhibitor ALW-II-41-27 suppresses EphA2 kinase activity in NSCLC cells by blocking the ATP-binding section of the kinase domain, with an IC50 value of 11 nM, reducing cell viability and triggering apoptosis in culture. When ALW-II-41-27 was injected into the abdominal cavity of mice with NSCLC tumour xenografts, tumour development was considerably reduced, whereas oral administration resulted in poor pharmacokinetic properties and limited oral bioavailability [76].

**Table 2 biomedicines-11-00315-t002:** Various small-molecule inhibitors/peptides/antibodies/ADCs to target Eph–ephrin receptors and their clinical trial status. ADC = Antibody–drug conjugate; ADCC = Antibody-dependent cellular cytotoxicity; IG = Immunoglobulin; MW = Molecular weight; N/A = Not applicable; VEGF = Vascular endothelial growth factor.

Drug	Type	Target	Mechanism of Action (MOA)	Clinical Trial	Clinicaltrials.gov Identifier, Reference
Azurin	Small-MW inhibitor	EphB2	Inhibits ligand binding, tyrosine phosphorylation	N/A	[107]
76D10	Small-MW inhibitor	EphA4	Inhibits ligand binding and tyrosine phosphorylation	N/A	[106]
Compound 1	Small-MW inhibitor	EphA4	Inhibits ligand binding, suppresses Akt phosphorylation, and induces apoptosis	N/A	[108]
NVP-BHG712	Kinase inhibitor	EphB4	inhibits EphB4 autophosphorylation and VEGF-driven vessel formation	N/A	[114]
Dasatinib	Kinase inhibitor	EphA2	Inhibits EphA2 phosphorylation, Cbl binding, internalisation, and degradation	Phase IV	NCT04155411 [115,118]
Compound 4a	Kinase inhibitor	EphA2	Blocks ATP access to the kinase and decreases cell viability of GBM cells	N/A	[117]
ALW-II-41-27	Kinase inhibitor	EphA2	Blocks ATP binding to the kinase domain	N/A	[76]
SWL	Peptide	EphA2	Induces EphA2 phosphorylation and blocks Erk/Akt pathways	N/A	[119]
SWL dimer	Peptide	EphA2	Induces EphA2 phosphorylation	N/A	[119]
SNEW	Peptide	EphB2	Blocks ephrin-B2 binding to EphB2	N/A	[120]
TNYL-RAW	Peptide	EphB4	Blocks ephrin-B2 binding to EphB4	N/A	[121]
DS-8895a	Antibody	EphA2	Antagonist, increases ADCC	Phase I	NCT02252211, NCT02004717 [122,123]
IG25	Antibody	EphA2	Induces EphA2 degradation	N/A	[124]
IG28	Antibody	EphA2	Inhibits ephrin-A1 binding to EphA2	N/A	[124]
EphA10/CD3	Antibody	EphA10	Induces tumour cell lysis and promotes T-cell activation	N/A	[125]
2H9	Antibody	EphB2	Induces internalisation of EphB2	N/A	[126]
131 and 47	Antibody	EphB4	Induces EphB4 degradation, inhibits tumour vasculature	N/A	[127]
IIIA4/KB004/Ifabotuzumab	Antibody	EphA3	Induces receptor phosphorylation and internalisation, and ADCC	Phase I	NCT03374943, [128,129,130]
IC1/MEDI-547	ADC	EphA2	Induces EphA2 phosphorylation, internalisation, and degradation	Phase I	NCT00796055, [131,132]
IIIA4-USAN	ADC	EphA3	Inhibits GBM cell viability/tumour growth	N/A	[43]
IIIA4-^177^Lu	Radio-labelled antibody	EphA3	Inhibits GBM cell viability/tumour growth	N/A	[43]
PF-06647263	ADC	Ephrin-A4	Binds to ephrin-A4-expressing cells and induces DNA cleavage and apoptosis	Phase I	NCT02078752, [133,134]

### 7.3. Peptides

Peptides have also been designed to bind Eph receptors, specifically targeting EphA2, EphB2, and EphB4, both as agonists and antagonists. The phage display method was used to discover the agonistic peptide SWL, which was demonstrated to activate EphA2 phosphorylation, and blocked major oncogenic pathways such as via Erk MAP kinases and Akt in PC3 prostate cancer cells, consistent with EphA2-mediated tumour suppression. While SWL had an IC50 of 4.1 µM, Duggineni et al. also generated an SWL dimer with 10-fold higher activity in inducing EphA2 tyrosine phosphorylation [119]. For EphB2, the peptide SNEW blocks binding to ephrin-B2 with an IC50 of about 15 µM by attaching to the receptor’s hydrophobic pocket [120]. Another antagonistic peptide is the 15-amino-acid-long TNYL-RAW peptide that specifically binds the ligand binding domain of EphB4, suppressing ephrin-B2 binding with an IC50 of 15 nM. TNYL-RAW’s affinity to the EphB4 relies heavily on the RAW sequence at its C terminus, and is the basis for further inhibitor development [121].

### 7.4. Antibodies

Monoclonal antibodies against several Ephs have also been generated, showing promising results [41]. Antibodies against the extracellular domain of EphA2 were produced by Carles-Kinch et al. and chosen for their ability to suppress the metastatic activity of breast cancer cells. EphA2 phosphorylation and degradation were both accelerated by treatment with this agonistic monoclonal antibody (mAb) [91]. The humanised anti-EphA2 mAb DS-8895a binds the extracellular juxtamembrane region of EphA2, and inhibits tumour development in EphA2-positive human breast and gastric cancer models, causing antibody-dependent cellular cytotoxicity (ADCC) [122]. The mAb DS-8895a has undergone evaluation in a Phase I study in Japanese patients with metastatic solid tumours. The antibody was well-tolerated up to 20 mg/kg, with only one out of thirty-seven patients discontinuing treatment owing to drug-related toxic effects [123]. Other EphA2 mAbs include antibody IG25, which promoted EphA2 degradation and reduced the growth of a pancreatic xenograft model, and IG28, which inhibited ephrin-A1 interaction, blocked tumour progression, and resulted in reduced tumour vascularisation when given to mice with orthotopic pancreatic tumours [124]. An EphA10 mAb was also developed which showed activity against breast tumour models by significantly suppressing their growth in a mouse xenograft model [125]. Among other anti-EphA10 monoclonal antibodies, clone #4 caused tumour regression and boosted the activation of CD8+ tumour-infiltrating cytotoxic T lymphocytes (CTLs) in vivo, while clone #9 triggered EphA10 internalisation in TNBC [135].

The anti-EphB2 mAb 2H9 (Genentech) was developed to antagonise EphB2–ephrin-B1 binding and induce the internalisation of nonphosphorylated EphB2 [126]. Vasgene (Los Angeles, CA, USA) and CNIO Biotechnology (Madrid, Spain) have both explored EphB4 and ephrinB2 as promising targets because of their crucial involvement in the formation of tumour vasculature. When combined with bevacizumab, the Vasgene mAbs 131 and 47 elicit tumour shrinkage in xenograft models via the degradation of EphB4, blocking tumour vasculature and slowing tumour progression [127].

The upregulation of EphA3 in a wide variety of solid and haematological cancers led to the development of the agonistic EphA3 mAb IIIA4 as an antitumor therapy. The IIIA4 mAb specifically binds EphA3+ tumour xenografts but not normal tissues [128]. IIIA4 is agonistic and causes EphA3 activation, the contraction of the cytoskeleton, and cell rounding in vitro. Naked IIIA4 also has antitumour activity in mouse xenografts, in which EphA3 is expressed either in the tumour cells or just in the tumour microenvironment [42]. IIIA4 treatment of EphA3-negative prostate cancer cell xenografts disrupted newly emerging tumour vessels and surrounding stroma, in line with EphA3 expression on these tissues in the TME. In a GBM mouse model, radiolabelled IIIA4 showed a substantial improvement in tumour inhibition, showing its efficacy as a tumour-targeting agent [41]. The human version of IIIA4, Ifabatuzumab (or KB004), targets EphA3 with a subnanomolar affinity. It was well-tolerated in a Phase 1 clinical study in haematological neoplasms, and showed clinical activity, particularly targeting the stromal/fibrotic tumour microenvironment in one patient [129]. A more recent Phase 1 clinical study for imaging radiolabelled IIIA4 in glioblastoma patients also showed the specific targeting of tumours with effects consistent with TME disruption [130].

### 7.5. Antibody–Drug Conjugates

Using antibodies that deliver cytotoxic payloads to specifically eliminate Eph-expressing tumour cells is another strategy to target Ephs as tumour antigens. Fourteen antibody–drug conjugates (ADCs) against other receptor targets have now advanced to the clinic, demonstrating the widespread acceptance of ADCs as a treatment modality [136]. ADCs include the direct conjugation of drugs to antibodies through noncleavable or cleavable linkers, with the latter being intended to aid drug release upon internalisation into tumour cells [41]. Maytansine (USAN) and monomethyl auristatin E (MMAE), also known as Vedotin, are two of the most widely used ADCs; both are strong antimitotic drugs that suppress division of cells by attaching to tubulin and disrupting microtubule formation [37]. Human monoclonal antibody 1C1 is an agonistic antibody against EphA2, causing rapid receptor phosphorylation, internalisation, and degradation. The ADC 1C1-maleimidocaproyl-MMAF (mcMMAF) triggered apoptosis of EphA2-expressing cells with an IC50 value of 3 ng/mL and suppressed tumour development in vivo. In mouse xenograft and rat syngeneic tumour models, application of 1C1-mcMMAF at 1 mg/kg showed a substantial growth suppression of EphA2-expressing tumours with no detectable negative impacts [131]. However, in a Phase I study (NCT00796055) of the anti-EphA2 1C1 auristatin conjugate MEDI-547, there were significant adverse events (AEs) at the starting dosage of 0.08 mg/kg, which included bleeding and blood coagulation, leading to the trial terminating [132].

An anti-EphA3 antibody–drug conjugate (ADC) based on the IIIA4 mAb attached to the microtubule antagonist maytansine (IIIA4-USAN) proved successful in eliminating GBM cells in vitro and inhibiting development of several GBM tumour types in mice compared to the naked antibody [43]. Likewise, the EphB2 mAb 2H9 proved ineffective against fibrosarcoma and colon cancer xenografts when used as a naked antibody but was able to reduce tumour cell growth when fused to the auristatin MMAE [126]. ADCs targeting ephrins have also been developed. Since ephrin-A4 was shown to be abundant in tumour-inducing or stem-like cells in triple-negative (TN) breast and ovarian patient-derived tissues, an ADC against ephrin-A4 was created using the humanised mAb E22 fused to the DNA-damaging agent calicheamicin. PF-06647263 successfully inhibited tumour development in xenograft models of TN breast and ovarian cancer [133].

## 8. Opportunities and Challenges for Therapeutic Targeting of Ephs and Ephrins

Eph receptors offer several potential approaches for developing novel cancer therapies, such as targets for monoclonal antibodies, peptides, and small-molecule Eph kinase inhibitors. Their enhanced expression in a wide range of tumour types, both in transformed cells and in the surrounding microenvironment, suggests exciting potential for development of Eph-targeted, tumour-selective therapies. However, such treatment strategies have challenges, such as inconsistent efficacy due to redundancy, the varying roles of Eph receptors on tumour growth, and the potential of harmful side effects due to expression in normal tissues [90]. While most active during embryonic development, some Ephs retain expression and roles in adult tissue homeostasis. Possible on-target cytotoxic effects of Eph receptor and ephrin treatments include effects on the cardiovascular system, bone homoeostasis, immunological activity, and neural function, all of which are controlled by Ephs/ephrins. Despite this, studies with Eph/ephrin-targeting drugs have not reported evident toxicity in mouse experiments [78]. While data in humans remains limited, naked antibodies against EphA2 and EphA3 were also well-tolerated in clinical trials, whereas ADCs targeting EphA2 and ephrin-A4 both caused significant adverse effects, and the trials were terminated [41]. This highlights the need for the careful selection of appropriately matched targets and therapeutic approaches.

A major limitation determining how to effectively target Eph receptors for clinical purposes is that the biology of the Eph/ephrin system is complex and still being defined [137]. Employing bidirectional signalling and crosstalk with other signalling pathways, Eph receptors and ephrins have significantly different activities in tumours dependent on the cellular and tissue context and the relative expression of binding partners. Interacting cells obtain mutually dependent signals from the same signalling complex, which may include multiple Eph family members that can crosstalk also with other receptor tyrosine kinases, as well as diverse downstream signalling pathways [37]. The same receptor, when overexpressed, can have both tumour-promoting and tumour-suppressing roles at different stages of cancer, as demonstrated in colon cancer [48]. Studies on EphA2 also show that levels of ligand-binding and receptor activity are critical, altering the balance between canonical tyrosine kinase signalling and noncanonical ligand-independent signalling, with the latter promoting a more invasive phenotype [23]. This is also consistent with observations that Eph receptors in tumours can show little or no apparent tyrosine phosphorylation [42,58], and suggests caution regarding the development of Eph kinase inhibitors.

Solving the problem of inhibiting harmful Eph activities while avoiding unwanted side effects will require careful analysis of Eph and ephrin expression profiles and activity in both tumours and normal tissues. Further examination of the consequences of Eph or ephrin depletion, enhanced expression/activity, and cancer-associated mutations in genetically manipulated mouse models that replicate the development of human tumours will also be important for improving our knowledge of Eph cancer biology. Both genetic and pharmacological targeting of the Eph system is best studied in vivo due to the complex expression profiles of Ephs/ephrins across diverse cell and tissue types [49]. Ideally, this should include immune-competent settings, allowing impacts on immune cell infiltration and activation in tumours to be assessed. Preclinical testing of possible combination therapies will also benefit from study of these mouse models to determine the effects of Eph expression and targeting on sensitisation to established therapies, as seen with trastuzumab resistance associated with EphA2 in HER2-positive breast cancer. Lastly, more studies of Eph and ephrin expression and activity in human tumours are required to evaluate Eph- and tumour-specific connections to disease progression, drug resistance, and patient survival, as well as identify other potential biomarkers of response. Together, these approaches can inform potential development of tumour- and patient-specific application of novel Eph-targeted therapies.

## 9. Conclusions

The re-emergence of the Eph tyrosine kinase receptors in a wide variety of malignancies indicates they play critical roles in tumourigenesis and are thus promising therapeutic targets. However, the differing expression and functions of Eph receptors across different types and stages of cancer indicates the need for detailed tumour-specific understanding to ensure the appropriate clinical use of Eph-targeted drugs. Disparities between tumour-suppressor and tumour-promoter actions are at least partly attributable to differences in ligand-dependent and ligand-independent signalling, as shown for EphA2 and EphB4. In particular, EphA2, EphA3, EphA4, and EphB4 are promising therapeutic candidates, based on increased expression in tumours and the TME, and drugs targeting these receptors have shown some promise in tumour models, with mixed success in the limited clinical studies performed to date. Future research will be critical to define the tumour-selective expression and function of this RTK family to in order to develop safe and effective Eph-targeted therapies.

## Figures and Tables

**Figure 1 biomedicines-11-00315-f001:**
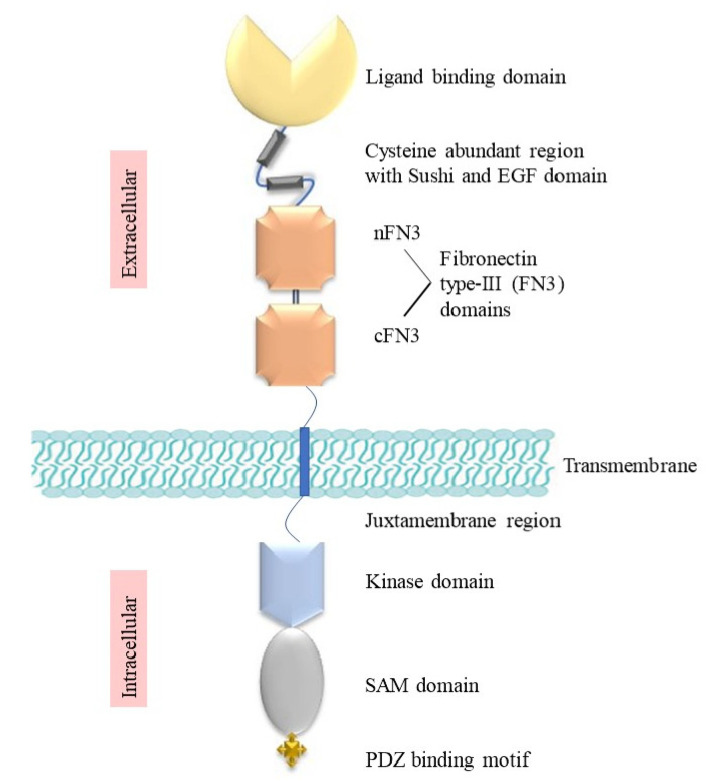
Structure of Eph receptors. Eph receptors are multidomain transmembrane proteins. The extracellular region consists of a ligand-binding domain (LBD) that binds to ephrin ligands on adjacent cells, a cysteine-rich domain (CRD) composed of a sushi and EGF domain, and two fibronectin III domains located C-terminally to the LBD. A juxtamembrane (JM) region, a kinase domain (KD), a SAM domain, and a C-terminal PDZ-domain-binding motif make up the Eph receptor cytoplasmic domain.

**Figure 3 biomedicines-11-00315-f003:**
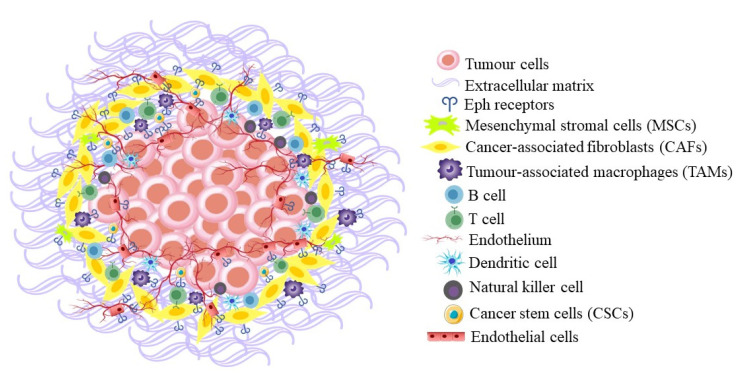
Eph receptors expressed on different cell types in the tumour microenvironment.

**Figure 4 biomedicines-11-00315-f004:**
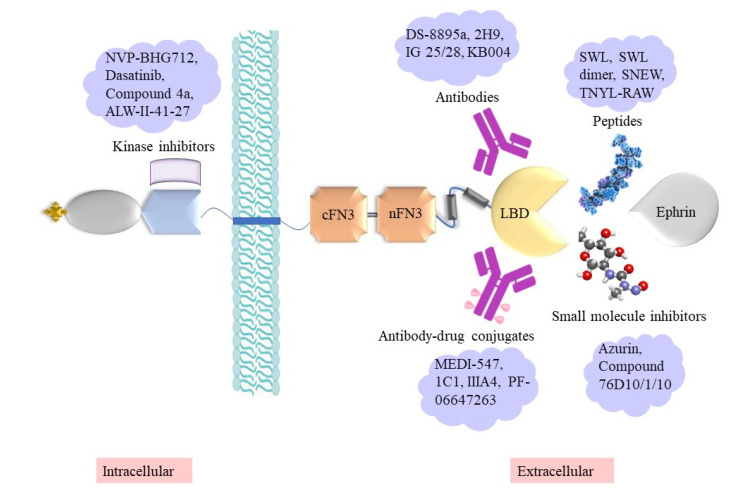
Illustration of potential therapeutic approaches targeting sites of Eph receptors.

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
