# Peer review of "Eph Receptors in Cancer"

_biomedicines, 2023, doi:10.3390/biomedicines11020315_

Round 1
Reviewer 1 Report
The review article entitled “Eph receptors in cancer” is a general overview related to the role of Eph tyrosine kinase receptors in a variety of malignancies. In addition, the authors report several strategies for the development of drugs that target Eph receptors in tumours, and their clinical and preclinical applications.
The overall quality of the present manuscript is good. I consider it to be of interest to the audience of Biomedicines, and I suggest its publication after the following minor issues are addressed.
Figure 2 should be modified to highlight that the ephrin ligand is placed on one cell, and the Eph receptor is located on an adjacent cell.
In the legend of Table 1 it would be necessary to add what the abbreviations used in the table stand for.
A final paragraph of conclusions should be added.
A list of abbreviations should be added before References section.
Many more recent references related to the therapeutic strategies to target Eph receptors need to be added.
Author Response
We thank the reviewers for their time and helpful comments, which we have now addressed as described below, with tracked changes shown in the revised manuscript.
Review 1
“Figure 2 should be modified to highlight that the ephrin ligand is placed on one cell, and the Eph receptor is located on an adjacent cell.” This figure has been modified to show ephrins on an adjacent cell membrane
“In the legend of Table 1 it would be necessary to add what the abbreviations used in the table stand for.” This has been done
“A final paragraph of conclusions should be added.” Conclusion section now added.
“A list of abbreviations should be added before References section.” Although this is not specified in the Instructions for authors, we have added a paragraph listing abbreviations, if deemed appropriate by the journal.
“Many more recent references related to the therapeutic strategies to target Eph receptors need to be added.” We have added more recent references, including some reviews covering specific topics in more depth.
Reviewer 2 Report
In this work titled “Eph receptors in cancer”, Dr. Arora and colleagues aim at describing the we discuss the expression and functions of Eph in a variety of malignancies and the multiple therapeutic in preclinical and clinical development.
The work is fairly up to date, well written. Figures are self-explanatory. I quite enjoyed reading it. I have some suggestions below which I fell would further strengthen the translational potential of this interesting work.
I would recommend having a specific paragraph addressing the general mechanism of resistance to therapy involving the Eph family, since this is a growing field of investigation in Eph biology.
At the same time, I would recommend more focus (if not a specific paragraph) on the prognostic value of Eph, which may deliver important insights into the heterogeneous contribution of Ephs in the various malignancies.
In general, it is very important to state whether the observed up/downregulation of Ephs has been observed at the mRNA or protein levels, since this may significantly change the interpretation of the results. For example, this is quiet unclear in the paragraph related to figure 3 (Ephs and tumor microenvironment).
Additionally , I would introduce more mechanistic explanation on the MOA of preclinical and clinical actionable compounds. As it is, it remains quite a schematic list which does not offer much to the reader. For example, an extra column may be added to the table 1, in place of the reference which may be much smaller.
Finally, there is previous work which should be mentioned because complementing the present manuscript, together with what the authors have already cited: for example doi: 10.3390/cancers13040700 ; doi: 10.3390/ijms232315275
Author Response
We thank the reviewers for their time and helpful comments, which we have now addressed as described below, with tracked changes shown in the revised manuscript.
Review 2
“I would recommend having a specific paragraph addressing the general mechanism of resistance to therapy involving the Eph family, since this is a growing field of investigation in Eph biology.”
We have added a paragraph specifically on this topic (new paragraph 5.5).
“At the same time, I would recommend more focus (if not a specific paragraph) on the prognostic value of Eph, which may deliver important insights into the heterogeneous contribution of Ephs in the various malignancies… In general, it is very important to state whether the observed up/downregulation of Ephs has been observed at the mRNA or protein levels, since this may significantly change the interpretation of the results. For example, this is quiet unclear in the paragraph related to figure 3 (Ephs and tumor microenvironment).”
We have added a new table (table 1) to include this information.
“Additionally , I would introduce more mechanistic explanation on the MOA of preclinical and clinical actionable compounds. As it is, it remains quite a schematic list which does not offer much to the reader. For example, an extra column may be added to the table 1, in place of the reference which may be much smaller.”
MOA information has been added, now in table 2.
“Finally, there is previous work which should be mentioned because complementing the present manuscript, together with what the authors have already cited: for example doi: 10.3390/cancers13040700 ; doi: 10.3390/ijms232315275”
We have added more recent references, including these reviews.